# Polarized lung inflammation and Tie2/angiopoietin-mediated endothelial dysfunction during severe *Orientia tsutsugamushi* infection

**Brandon Trent[1], Yuejin Liang[2], Yan Xing[2¤a], Marisol Esqueda🔳[2], Yang Wei[2¤b], Nam-Hyuk Cho[3,4], Hong-II Kim[4], Yeon-Sook Kim[5], Thomas R. Shelite[2¤c], Jiyang Cai[6¤d], Jiaren Sun[1,2], Donald H. Bouyer[1], Jinjun Liu[7]\*, Lynn Soong🔳[1,2]\***

1 Department of Pathology, University of Texas Medical Branch, Galveston, Texas, United States of America, 2 Department of Microbiology and Immunology, University of Texas Medical Branch, Galveston, Texas, United States of America, 3 Department of Microbiology and Immunology, Seoul National University College of Medicine, Seoul Republic of Korea, 4 Department of Biomedical Sciences, Seoul National University College of Medicine, Seoul, Republic of Korea, 5 Division of Infectious Diseases, Department of Internal Medicine, Chungnam National University School of Medicine, Daejeon, Republic of Korea, 6 Department of Ophthalmology & Visual Sciences, University of Texas Medical Branch, Galveston, Galveston, Texas, United States of America, 7 Department of Physiology and Pathophysiology, School of Basic Medical Sciences, Xi'an Jiaotong University Health Science Center, Xi'an, China

¤a Current address: People's Hospital of Henan Province, Henan, China
¤b Current address: Core Research Laboratory, The Second Affiliated Hospital of Xi'an Jiaotong University, Xi'an, China
¤c Current address: Department of Internal Medicine, Division of Infectious Diseases, University of Texas Medical Branch, Galveston, Texas, United States of America
¤d Current address: Department of Physiology, OUHSC, Oklahoma City, Oklahoma, United States of America
\* jupet@163.com (JL); lysoong@utmb.edu (LS)

**Data Availability Statement:** All relevant data are within the manuscript and its Supporting Information files.

## Abstract

*Orientia tsutsugamushi* infection can cause acute lung injury and high mortality in humans; however, the underlying mechanisms are unclear. Here, we tested a hypothesis that dysregulated pulmonary inflammation and Tie2-mediated endothelial malfunction contribute to lung damage. Using a murine model of lethal *O. tsutsugamushi* infection, we demonstrated pathological characteristics of vascular activation and tissue damage: 1) a significant increase of ICAM-1 and angiopoietin-2 (Ang2) proteins in inflamed tissues and lung-derived endothelial cells (EC), 2) a progressive loss of endothelial quiescent and junction proteins (Ang1, VE-cadherin/CD144, occuludin), and 3) a profound impairment of Tie2 receptor at the transcriptional and functional levels. *In vitro* infection of primary human EC cultures and serum Ang2 proteins in scrub typhus patients support our animal studies, implying endothelial dysfunction in severe scrub typhus. Flow cytometric analyses of lung-recovered cells further revealed that pulmonary macrophages (MΦ) were polarized toward an M1-like phenotype (CD80+CD64+CD11b+Ly6G-) during the onset of disease and prior to host death, which correlated with the significant loss of CD31+CD45- ECs and M2-like (CD206+CD64+CD11b+Ly6G-) cells. In vitro studies indicated extensive bacterial replication in M2-type, but not M1-type, MΦs, implying the protective and pathogenic roles of M1-skewed responses. This is the first detailed investigation of lung cellular

**Funding:** This work was supported by generous grant funding from the NIH and NIAID (grant numbers AI132674, AI126343 to LS, https://www.niaid.nih.gov/), a T32 Biodefense Training Program Grant (grant number AI060549, provided to BT, https://www.niaid.nih.gov/), and a T35 Infectious Diseases Training Grant (grant number AI078878 to LS and ME, https://www.niaid.nih.gov/). Funding was also provided from the UTMB Infection and Immunity Data Generation Award to LS (https://www.utmb.edu/ihii). Lastly, funding was provided by the American Association of Immunologists as a 2018 AAI Careers in Immunology Fellowship (grant to BT and LS, https://www.aai.org/). The funders had no role in study design, data collection and analysis, decision to publish, or preparation of the manuscript.

**Competing interests:** The authors have declared that no competing interests exist.

immune responses during acute *O. tsutsugamushi* infection. It uncovers specific biomarkers for vascular dysfunction and M1-skewed inflammatory responses, highlighting future therapeutic research for the control of this neglected tropical disease.

## Author summary

Scrub typhus is a life-threatening disease, infecting an estimated one million people yearly. Acute lung injury is a dangerous clinical development in severe cases; however, its pathogenic biomarkers and mechanisms of progression remain unknown. Here, we used a lethal infection mouse model that parallels certain aspects of severe scrub typhus, primary human endothelial cell cultures, and patient sera to define pathogenic biomarkers following *Orientia tsutsugamushi* infections. We found a significant increase in the levels of endothelial activation/stress markers (angiopoietins and ICAM-1) in infected mouse lungs and patient sera, but a progressive loss of endothelium-specific Tie2 receptor and junction proteins (VE-cadherin), at severe stages of disease. These signs of vasculature disruption positively correlated with the timing and magnitude of recruitment/activation of proinflammatory MΦ subsets in infected lungs. Bacterial growth *in vitro* was favored in M2-like, but not in M1-like, MΦs. This study, for the first time, reveals endothelial malfunction and dysregulated inflammatory responses, suggesting potential therapeutic targets to ameliorate tissue damage and pathogenesis.

## Introduction

Scrub typhus is a febrile and potentially lethal illness that infects an estimated one million individuals per year [1]. The disease is caused by infection with the bacterium, *Orientia tsutsugamushi*. Nearly a third of the human population lives in endemic areas, known as the "tsutsugamushi triangle", although recent reports have identified scrub typhus in South America, which was previously believed to be free of scrub typhus [1, 2]. Within endemic areas, scrub typhus is reported to cause a substantial proportion (approximately 15–23%) of reported febrile illness [3, 4]. If left untreated, scrub typhus can manifest as interstitial pneumonia, myocardial and hepatic inflammation, and meningoencephalitis. [5]. Mild interstitial pneumonitis is typically the extent of pulmonary involvement during self-resolving or promptly treated scrub typhus. However, life-threatening pathologies can arise in severe cases, including lung hemorrhage, edema buildup, diffuse alveolar damage, and interstitial cellular infiltration [6]. Acute respiratory distress syndrome and lung damage are associated with high mortality and present in 6.75–25% of scrub typhus patients [5, 7]; however, there is no detailed investigation of the underlying mechanisms responsible for pulmonary endothelial dysfunction and inflammation.

Being an obligate intracellular bacterium, *O. tsutsugamushi* can infect a variety of host cells but primarily replicate in macrophages (MΦs)[8], dendritic cells [9], and endothelial cells (ECs) [6, 8]. The bacteria enter host cells via the phagosome [10] or endosome [11], which they subsequently escape to begin replication in the cytoplasm. Infection-triggered cellular responses, including the activation of activator protein-1 (AP-1) and NF-κB pathways, the production of proinflammatory cytokines (IL-1β, TNF-α, IL-8/CXCL8), and the expression of distinct gene profiles, have been examined *in vitro* by using primary human umbilical vein endothelial cells (HUVEC) [12], human epithelial/EC-like ECV304 cell line [13], human

monocytes or MΦs [14]. Prolonged infection can result in EC death via apoptosis [15], but there is limited information on endothelial responses during the course of *O. tsutsugamushi* infection [16]. Staining of *O. tsutsugamushi* antigen in both human scrub typhus cases and murine infection models has identified pulmonary endothelium as a location of bacterial tropism [8, 17, 18]. Sublethal *O. tsutsugamushi* infection studies in outbred Swiss CD-1 mice [19], as well as clinical studies of human patients [20], have shown significant elevation of endothelial activation markers (ICAM-1, VCAM-1, E-Selectin, etc.) in the serum of infected individuals; however, endothelium-focused analyses during *in vivo* infection remain largely unexplored [21]. Several studies have characterized the response and activation of MΦs during *O. tsutsugamushi* infection [22, 23]. *In vitro* and *ex vivo* experiments have shown that human monocytes/MΦs in the circulation adopt an inflammatory "M1" type transcriptional profile after *O. tsutsugamushi* infection, although little is known regarding tissue specific macrophages or the presence of alternatively activated "M2"-type macrophages [24].

The delicate balance of EC quiescence and activation is crucial during systemic infection. While EC activation promotes adherence and recruitment of innate and adaptive immune cells for pathogen clearance, prolonged activation can lead to EC cytotoxicity, impaired barrier function, and host mortality [25]. One of the critical mechanisms to control EC activation status and cellular function is through competitive interactions between angiopoietin-1 (Ang1) and Ang2 ligands with their receptor, Tie2, a protein tyrosine kinase that is predominately expressed on ECs in humans and mice. Tie2 activation and phosphorylation via binding with Ang1 (produced by pericytes and platelets [26]) promote EC quiescence, which limits leukocyte adhesion and maintains EC survival and vascular barrier integrity [27]. Infection- or inflammation-triggered release of Ang2 (normally stored within the Weibel-Palade bodies in ECs [28, 29]) can compete with Ang1 binding Tie2 to antagonize its function [30]. Inhibition of Tie2 signaling via Ang2 binding stimulates leukocyte adhesion, vascular barrier destabilization, and inflammation [31, 32]. Thus, Ang2/Ang1 expression ratios and Tie2 activation status are important biomarkers for the pathogenesis of systemic infection, such as severe sepsis and malaria [33, 34].

To investigate endothelial alterations during severe *O. tsutsugamushi* infection, we have recently developed a lethal intravenous (i.v.) *O. tsutsugamushi* infection model in C57BL/6 mice [35–37], which parallels aspects of severe scrub typhus in humans. In our lethal models, bacterial loads in both the spleen and liver reached peak levels around or shortly after the onset of disease (days 6–8 post-infection), but are reduced significantly by days 10–12. In contrast, lung bacterial loads remain elevated throughout infection [17]. All mice expire by days 12–13, suggesting unknown mechanisms of pathogenesis are at work during late infection [17, 35]. Given that the lungs are a major organ for *O. tsutsugamushi* infection in humans and in different animal models, and that elevated ratios of *ANG2/ANG1* transcripts are pathological hallmarks in lethal infection models [17, 35], we hypothesize that dysregulated pulmonary inflammation and Tie2/Ang2-mediated endothelial dysfunction contribute to disease pathogenesis at late stages of *O. tsutsugamushi* infection.

In this study, we utilized a lethal infection model in C57BL/6 mice, *in vitro* infection systems (human EC cultures, and bone marrow-derived MΦ subsets), as well as sera from scrub typhus patients to reveal positive correlations between vascular dysfunction, activation of innate immune cells, and disease progression. We focused on two cellular subsets known to be sites of *O. tsutsugamushi* replication, ECs and MΦs, to characterize their activation and polarization *in vivo*. To the best of our knowledge, this is the first report to delineate MΦ subsets in inflamed lungs and their positive correlation with Tie2 malfunction during late stages of severe infection.

## Results

### Pulmonary EC activation and tight junction disruption during infection in mice

Given that the lung is a major site of *O. tsutsugamushi* infection in humans and animal models [6, 35] and that EC activation and disruption of vascular barrier integrity are principal steps for acute lung injury in sepsis and pneumonia models [38], we sought to investigate pulmonary EC activation in C57BL/6 mice following i.v. inoculation with a lethal dose of *O. tsutsugamushi* Karp (~$1.325 \times 10^6$ viable bacteria in 200 µl of PBS). Inoculation via this route establishes bacterial replication in the lungs accompanied by interstitial pneumonitis and alveolar thickening (S1A Fig, [17]). Immunofluorescent staining of frozen lung sections revealed increased ICAM-1-positive (green) vascular staining on days 2 (D2) to 9, which correlated with the increase of detected bacterial antigen (red) (Fig 1A, boxed area, Fig 1C). To examine endothelial structure and adherens junctions, we co-stained lung sections with GSL I-B$_4$ lectin (specific for $\alpha$-galactose residues known to be enriched on the surface of EC) and anti-VE-cadherin/CD144 (an adherens junction protein, red), as described in our previous report for neuroinflammation [37]. The VE-cadherin staining was intense and homogenous in the control tissues, but markedly reduced in D2 samples; VE-cadherin staining was nearly absent in some foci of D6 and D9 samples (Fig 1B and 1C), implying the reduction of junction proteins. In conjunction with the endothelial junction proteins, we co-stained the epithelial junction protein, occludin, and GSL I-B$_4$ lectin. Consistently, we found a progressive loss of occludin staining on the bronchial epithelium during the infection and a near absence of staining in D9 samples (S1B Fig). These data suggest a progressive and severe loss of vascular barrier integrity in the infected lungs, especially at late stages of acute infection prior to host death (D9) [35].

To support our immunofluorescent results, we prepared single-cell suspensions from mouse lung tissues and used flow cytometry to examine the frequency of recovered CD31$^+$CD45$^-$ ECs and their surface ICAM-1 expression. We found that compared with the mock controls, infected lung tissues contained a significant increase in the frequencies of ICAM-1$^+$CD31$^+$CD45$^-$ ECs at D6 and D9, respectively ($p < 0.001$, $p < 0.01$, Fig 1D), while there were approximately 5-folds reduction in the frequencies of total ECs at D6 and D9 ($p < 0.001$, S1C Fig). Flow cytometry data reinforced the immunofluorescent results, indicating marked endothelial activation and damage at D6 (the onset of disease) and D9 (prior to host demise). To validate these findings in mice, we infected primary HUVEC cultures with different doses of *O. tsutsugamushi* (3 and 10 MOI) and found a dose-dependent increase in *ICAM1* and *IL8/CXCL8* transcripts at 24 h post-infection (S1D Fig). Collectively, these data indicate infection-triggered endothelial stress and activation, accompanied by progressive vascular damage and tight junction disruption during the course of infection.

### Alterations in the angiopoietin-Tie2 system during *Orientia* infection

Currently, there are no detailed *in vivo* studies to define molecular mechanisms underlying *O. tsutsugamushi* infection-associated vascular damage. For other severe and systemic infections caused by bacteria or viruses, alterations in angiopoietin proteins or their functional Tie2 receptor is considered as one of the key mechanisms for vascular dysfunction [39, 40]. Given our previous findings of elevated *ANG2/ANG1* mRNA ratios in mice with severe scrub typhus and in *O. tsutsugamushi*-infected HUVECs [36], we speculated that impairment in Tie2 function occurs in severe scrub typhus [16]. To test this hypothesis, we examined Ang1, Ang2, and Tie2 protein levels in the lung tissues via immunofluorescent staining (Figs 2 and 3). While the mock controls contained relatively high levels of Ang1 (green), with relatively low levels of

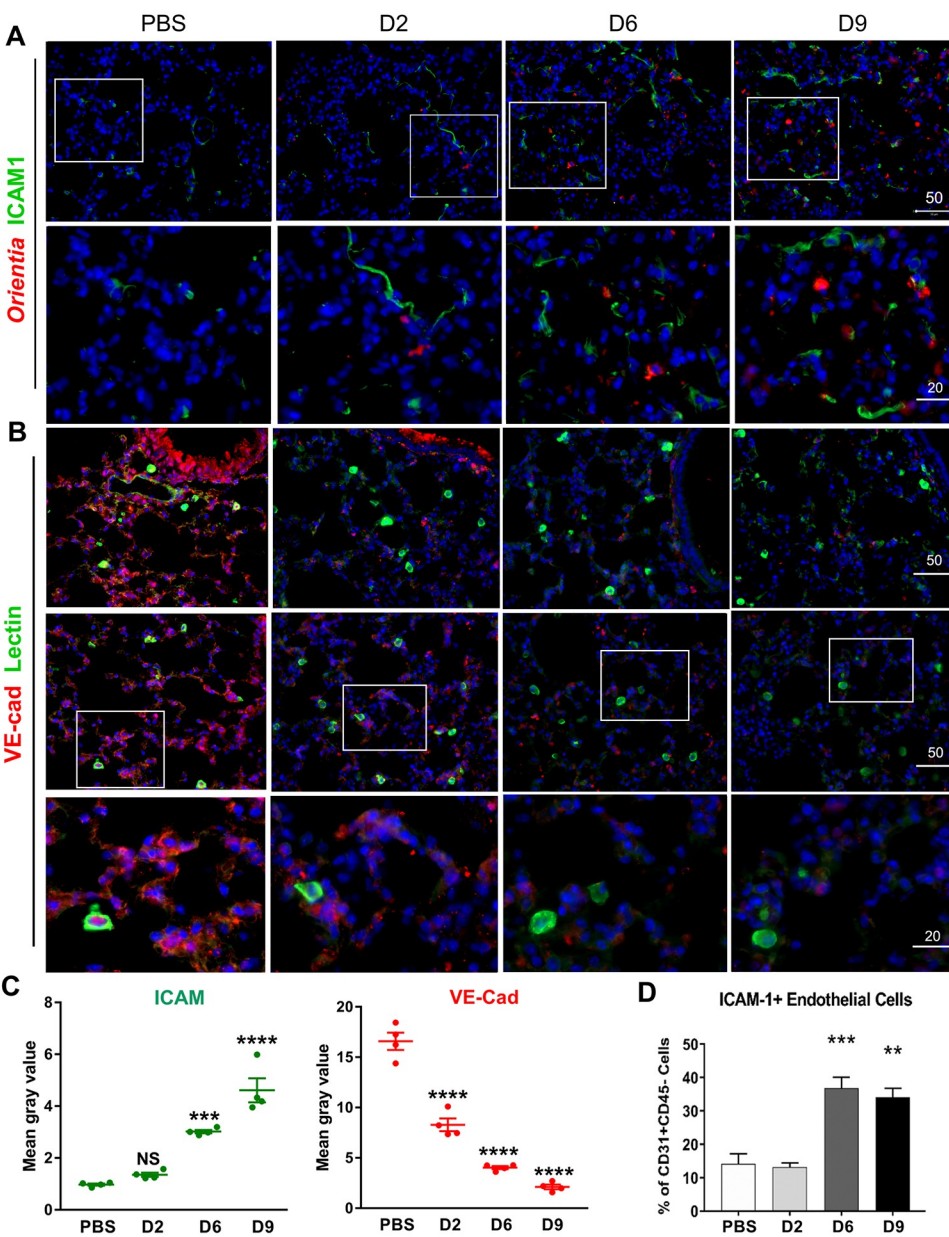

**Fig 1. Endothelial cell (EC) activation and vascular damage in the lungs of *O. tsutsugamushi*-infected mice.** Female C57BL/6J mice were inoculated with 1.325 x 10^6 of *O. tsutsugamushi* Karp strain (4–5 mice/group) or PBS (3–4 mice/group). At days 2, 6, or 9 post-infection, equivalent lung portions were collected for analyses. (A) Frozen lung sections were co-stained for *Orientia* bacteria (red), ICAM-1 (green), and DAPI (blue, top row, scale bar = 50 μm) with close-up views of the boxed areas in the bottom row (bar = 20 μm). (B) Lung sections were co-stained for VE-cadherin (adherens junctions, red), FITC-labeled I-B$_4$ lectins (green), and DAPI (blue, top row, bars = 50 μm). Close-up views of the boxed areas located the bottom row (bar = 20 μm). (C) Quantification of fluorescent ICAM-1 and VE-Cadherin staining (four images per time point). (D) Lung-derived cells were analyzed via flow cytometry for the percentage of ICAM-1$^+$ cells among gated CD31$^+$CD45$^-$ ECs (4–5 mice/group in infected groups; 3 mice/group in PBS groups). $^*$, $p < 0.05$; $^{**}$, $p < 0.01$; $^{***}$, $p < 0.001$; and $^{****}$, $p < 0001$ compared to PBS controls. Graphs are shown as mean +/- SEM. Flow cytometric and qRT-PCR data were analyzed by using one-way ANOVA with Tukey's Post Hoc. At least 4 independent mouse infection experiments and 2 independent *in vitro* experiments were performed with similar trends; shown are representative data.

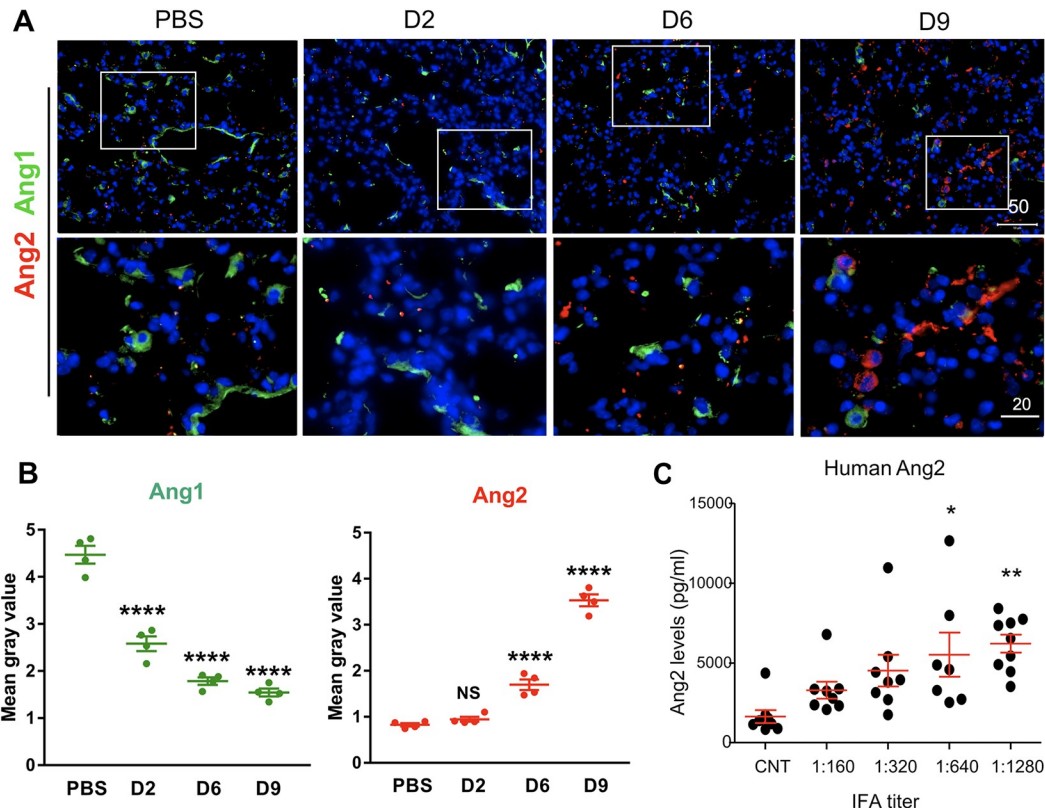

**Fig 2. Elevated Ang2 expression and decreased Ang1 expression during *O. tsutsugamushi* infection.** Mice were infected as in *Fig 1*. (A) Frozen lung sections were co-stained for Ang1 (a marker for endothelial quiescence, green), Ang2 (an endothelial stress marker, red), and DAPI (blue) showing images at a low magnification (top row, scale bar = 50 μm) and close-up views of the boxed areas (bottom row, bar = 20 μm). (B) Quantification of fluorescent Ang1 and Ang2 staining (four images per time point). (C) Human serum Ang2 proteins in the control subjects (CNT) or scrub typhus patients (8/group) with different anti-*Orientia* IFA antibody titers were measured by ELISA. Shown are data from two independent experiments. *, $p < 0.05$; **, $p < 0.01$ and ***; $p < 0.001$; ****; $p < 0001$ compared to the controls. Graphs are shown as mean +/- SEM. Serum ELISA and qRT-PCR groups were analyzed via one-way ANOVA with Tukey's Post Hoc. At least 3 independent mouse infection experiments were performed with similar trends; shown are representative data.

Ang2 (red), we found a modest decrease in Ang1-positive staining but a steady increase in Ang2 staining, during the infection (Fig 2A). These IFA results were quantified, demonstrating the significant decrease of Ang1 and a significant increase of Ang2 expression during *O. tsutsugamushi* infection that correlated with disease progression (Fig 2B). Increases in Ang2 staining were also accompanied by increasing levels of markers of neutrophil activation, such as myeloperoxidase (MPO, S2 Fig). To validate our findings mouse models, we measured scrub typhus patient sera via specific ELISA assays and found a statistically significant increase in circulating Ang2 levels, which correlated with their *O. tsutsugamushi*-specific antibody titers ($p < 0.05$ and $p < 0.01$, comparing IFA titers of 1:640 and 1:1280 with the control subjects, Fig 2C). These human data support our findings obtained from mouse tissues, indicating the potential utility of serum Ang2 levels as a molecular biomarker of scrub typhus severity.

IFA staining of the Ang1/2 receptor, Tie2, revealed clear Tie2 staining in mock infected controls; however, positive Tie2 staining was nearly absent in some foci of D6 and D9 samples (Fig 3A). To validate these findings, we used Western blot analyses of lung tissues and confirmed a striking reduction of phosphorylated Tie2 (pTie2) and total Tie2 levels at both D6 and D9, as compared with either the mock and D2 samples (Fig 3B), implying impairments at

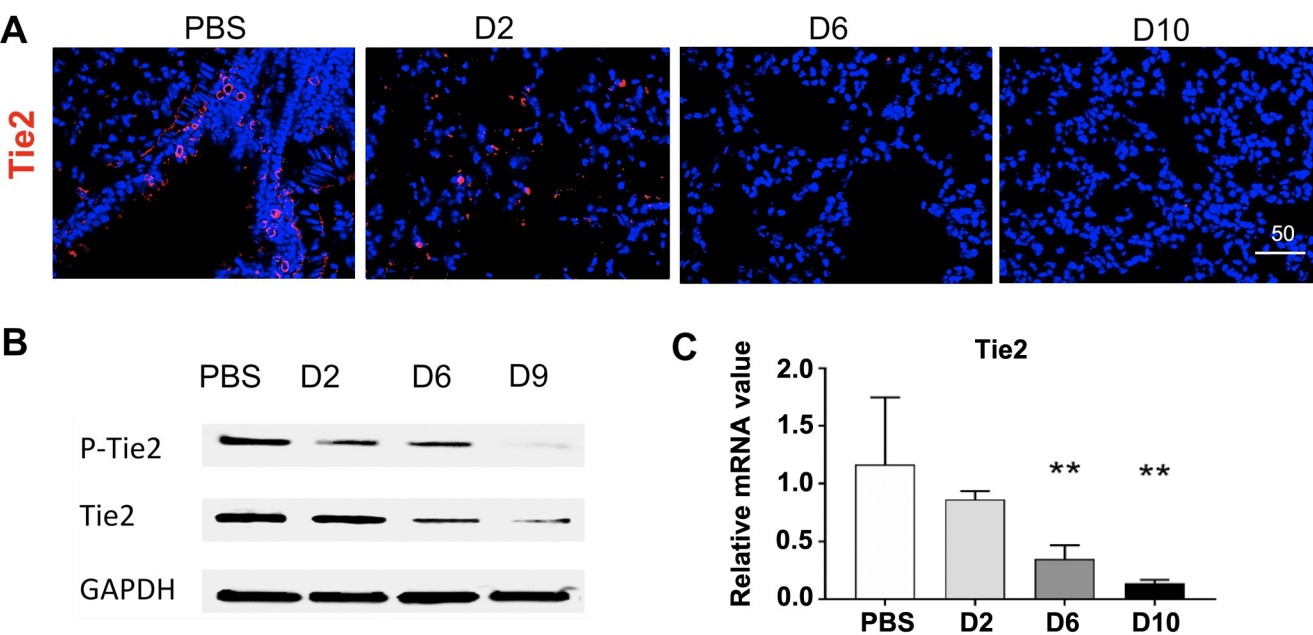

**Fig 3. Reduced Tie2 expression and activation in the lungs of *O. tsutsugamushi* infected mice. (A)** Frozen lung tissue sections were stained for the Tie2 receptor (red) and DAPI (blue, bar = 50 μm). (B) Lung tissue homogenates (40 μg/lane) were measured by Western blots for the levels of phospho-Tie2 (pTie2) and total Tie2 proteins and compared with the GAPDH controls. (C) *TIE2* mRNA levels in mouse lungs were measured via qRT-PCR; data are presented as relative mRNA values normalized to β-actin. **, $p < 0.01$ compared to the controls. Graph shown as mean +/- SEM. Serum ELISA and qRT-PCR groups were analyzed via one-way ANOVA with Tukey's Post Hoc.

the translational and functional levels. The qRT-PCR analyses further validated a statistically significant decrease in *TIE2* mRNA levels in the lungs at D6 and D10 ($p < 0.01$, compared with the mock controls, Fig 3C), implying impairment at the transcriptional level. These data, together with our previous studies [35, 36], indicate that marked Ang2 production, accompanied with severe impairment in the Tie2 functions, are pathogenic mechanisms of severe vascular damage in *O. tsutsugamushi* infection.

## M1-like responses in the lungs of infected mice

Having documented progressive endothelial damage and alterations in endothelium-specific biomarkers following *O. tsutsugamushi* infection (Figs 1 and 2), we then examined the timing and magnitude of leukocyte recruitment and activation. Although some reports described leukocyte involvement in *O. tsutsugamushi*-infected mouse spleen and brain [37, 41], there are no detailed studies of innate immune responses in infected lungs. Using immunofluorescent staining, we found that CD45[+] leukocytes and CD3[+] T cells were accumulated around Ang2-positive foci in the lungs at D6 and D10, and that CD45-Ang2 or CD3-Ang2 co-stained foci were readily detectable at D10 (yellow, S3A Fig). Flow cytometric analyses revealed a 20-fold increase in total numbers of CD4[+] T cells at D10, but a statistically significant decrease in percentages of these cells at D6 and D10, respectively ($p < 0.0001$, compared with mock controls, S3B Fig). In contrast, there was a 50-fold increase in total numbers and 2.3-fold increase in percentages of CD8[+] T cells at D10 (S3C Fig). These findings were consistent with the known importance of CD8[+] T cells during *O. tsutsugamushi* infection in mice [41, 42].

Monocytes and MΦs are particularly noteworthy leukocytes during *O. tsutsugamushi* infection, as they can act as a target for bacterial replication and a propagator of the inflammatory

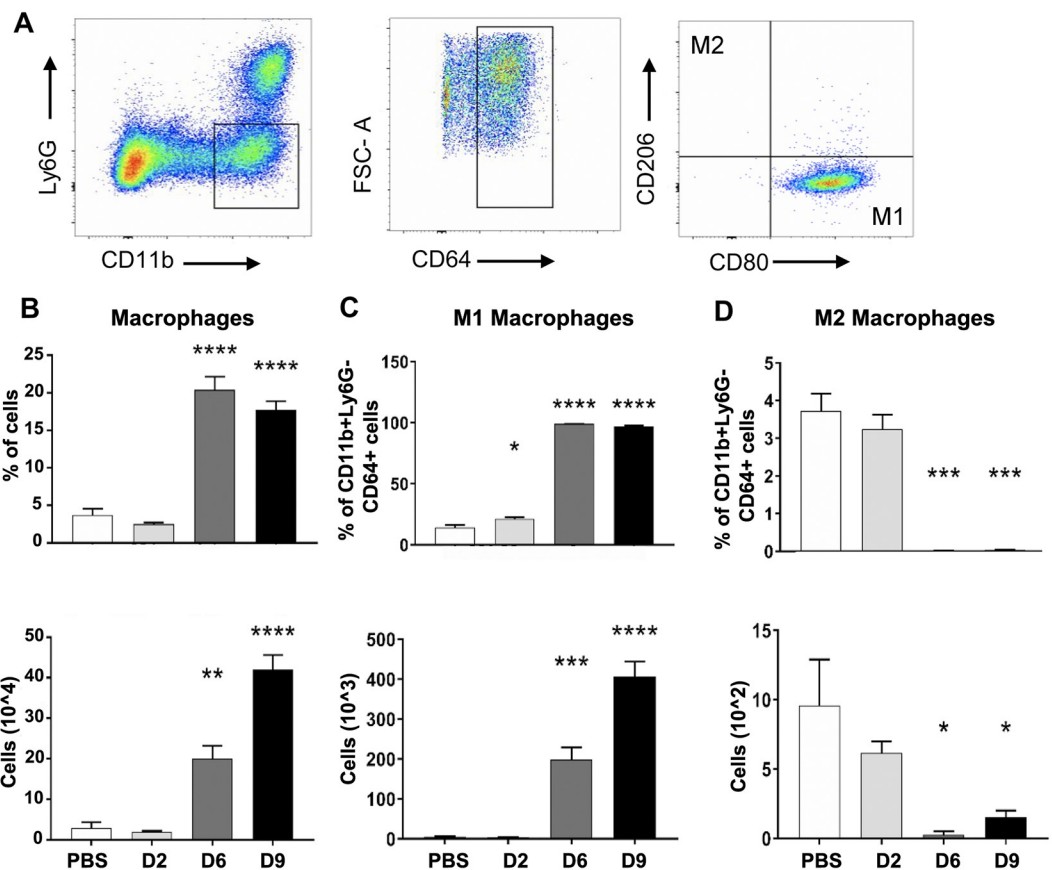

**Fig 4. Polarized MΦ activation in infected mouse lungs.** Mice were infected with *O. tsutsugamushi* (4–5 mice/group) or PBS (3–4 mice/group) for lung tissues collection at indicated days of infection, as in *Fig 1*. (A) Flow cytometric analyses of lung-derived cells, gated on CD11b+Ly6G- MΦs and MΦ subsets, are shown for the D9 samples. The percentages and total numbers of (B) MΦs (CD64+CD11b+Ly6G-), (C) M1-type MΦs (CD80+CD206- CD64+CD11b+Ly6G-), as well as (D) M2-type MΦs (CD206+CD80- CD64+CD11b+Ly6G-) are shown, respectively. *, $p < 0.05$; **, $p < 0.01$; ***, $p < 0.001$, ****, $p$ 0.0001 compared to the PBS controls.

response [8, 14], possibly playing a role in *O. tsutsugamushi* dissemination from skin lesions [9, 43]. While *in vitro* infection predominantly drives human monocytes/MΦs to M1-like transcription programs [9, 43], our current knowledge on *O. tsutsugamushi*-MΦ interactions in the lungs is still limited. Using IFA staining, we observed co-localization of bacteria (green) with IBA-1+ MΦs (red) in mouse lungs (S4A Fig). To define monocyte/MΦ responses, we applied reported protocols and gating strategies for flow cytometric analysis of mouse lung monocyte/MΦ subsets [44] (Fig 4A). Compared with the mock controls, D6 and D9 samples had 4- to 5-fold increases in the frequency of CD64+CD11b+Ly6G- alveolar/interstitial monocytes/MΦs, as well as 6- and 14-fold increases in total cell numbers, respectively ($p < 0.01$ and $p < 0.001$, Fig 4B). Using previously reported markers for M1-kike MΦs (CD80+CD206- CD64+CD11b+Ly6G-) or M2-like MΦs (CD80-CD206+CD64+CD11b+Ly6G-) [45], we found that nearly all (~97%) pulmonary MΦs displayed an M1-like phenotype at D9 (Fig 4C). In contrast, while the mock and D2 lung samples contained ~3.2% of M2-like cells, these cells were barely detectable at D6 or D9 (Fig 4D). These data suggest extensive recruitment and/or activation of M1-like cells, but a marked loss and/or suppression of M2-like cells, during the progression of disease. Likewise, lung qRT-PCR assays confirmed a significant up-regulation of

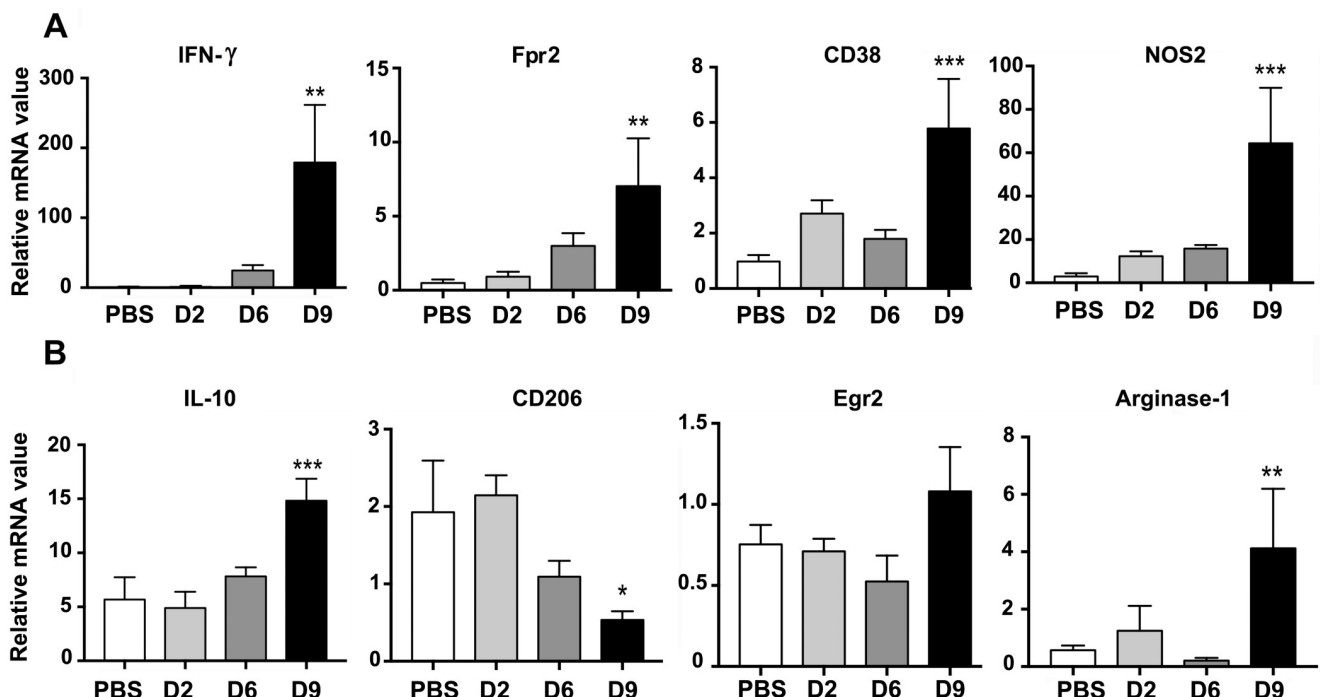

**Fig 5. Transcription of M1 and M2 associated genes in the lung of infected mice.** Lung tissues were measured for the expression of M1-related genes (A) and M2-related genes (B), respectively. Data are presented as relative to β-actin values. $^*$, $p < 0.05$; $^{**}$, $p < 0.01$; $^{***}$, $p < 0.001$ compared to the PBS controls. Graphs are shown as mean +/- SEM. One-way ANOVA with Tukey's Post Hoc was used for statistical analysis. Two independent mouse infection experiments were performed with similar trends; shown are representative data.

M1 markers (*IFNγ*, *FPR2*, *CD38*, and *NOS2*), but not M2 markers (*CD206*, *EGR2*), at D9 (Fig 5A and 5B). While the *IL10* up-expression was previously reported by our lab and other groups [46], we also detected an increased expression of *ARGINASE1* (Fig 5B), a marker known for M2 polarization and the growth of other intracellular pathogens [47]. Together with data shown in Fig 4, we concluded that at the onset of disease and beyond, *O. tsutsugamushi* infection preferentially stimulated pro-inflammatory innate responses in M1-like monocytes/MΦs, which correlate with the onset of vascular damage (Figs 1–3).

## MΦ polarization in favor of *Orientia* replication *in vitro*

Because we had demonstrated differential monocyte/MΦ responses *in vivo*, we asked how MΦ polarization might influence intracellular growth of the bacteria. We generated bone marrow-derived MΦs from naïve C57BL/6 mice, polarized cells via pretreatment with LPS (100 ng/ml), recombinant IL-4 (rIL-4, 10 ng/ml), or no treatment, for 24 h, infected cells with *O. tsutsugamushi* (MOI 5), and measured bacterial loads at different time points. Flow cytometry and gene profile analyses of primed but uninfected cells confirmed their corresponding polarization to either classically activated M1 or alternatively activated M2 phenotypes (S4B and S4C Fig), as documented by others [48]. At 48 h post-infection, both IL-4-primed M2 cells and non-polarized M0 cells contained significantly increased loads of bacteria (determined by the copy number of *Orientia* 47-kDa gene) than LPS-primed M1 cells ($p < 0.0001$, Fig 6A). At 72 h post-infection, M2 cells contained 10-fold more bacteria than M1 cells (Fig 6A), with extensive accumulation of bacteria (green) within IBA-1-positive MΦs (red, Fig 6B). M2 polarized cells also contained significantly increased bacterial loads compared to M0 cells at 72 hpi, although M0 cells still carried far greater numbers of bacteria than M1 polarized cells (Fig 6A).

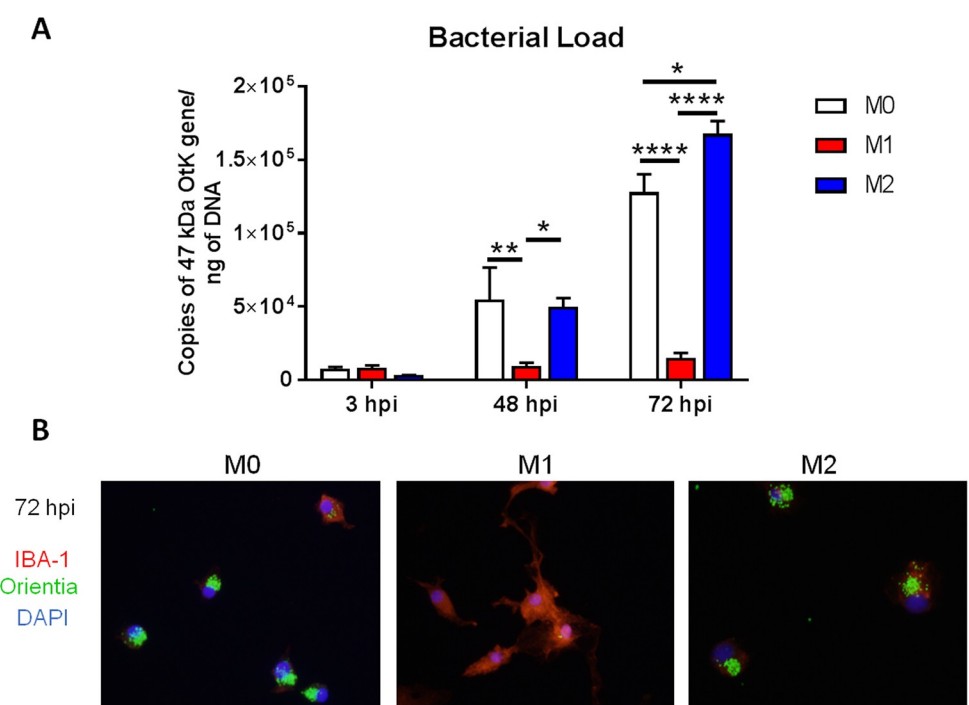

**Fig 6. Enhanced bacterial growth in M2-polarized MΦs.** Bone marrow-derived MΦs were generated from C57BL/6J mice, polarized into M1 or M2 types by pre-treatment of cells with LPS (100 ng/ml) or rIL-4 (10 ng/ml), or given media only for non-polarized M0 MΦs. MΦs were then infected with bacteria (5 MOI). (A) Bacterial loads at 3, 48, and 72 hpi (n = 5) were determined by qPCR. Data are presented as the *Orientia* 47-kDa gene copy per pg of DNA. (B) Cells were co-stained for *Orientia* (green), IBA-1 (a MΦ marker, red), and DAPI (blue) at 72 hpi. *, $p < 0.05$; **, $p < 0.01$; ****; $p < 0001$. Data for qPCR was analyzed using two-way ANOVA with Tukey multiple comparison test.

## Discussion

Despite being an important emerging infectious disease, detailed immunological studies of scrub typhus patient samples or *O. tsutsugamushi*-infected animal tissues during the course of disease are scarce. In this study, we focused on ECs and monocytes/MΦs in the lungs of lethally infected mice to examine the activation of these cellular subsets known to be important cellular targets of *O. tsutsugamushi in vivo*. Our findings revealed important parameters and cell-specific alterations associated with acute lung injury and pathogenesis. The endothelium in infected lungs presented progressive Tie2 malfunction, increased Ang2 and ICAM-1 expression and pro-inflammatory MΦs at the onset of disease and severe stages of infection. Since lung damage and vascular malfunction are hallmarks of scrub typhus severity in patients [6], a better understanding of pathogenesis associated with acute lung injury is important for disease control and management.

The molecular characteristics of endothelial alterations during *O. tsutsugamushi* infection *in vivo* have not been explored. The present study provided evidence for the potential mechanisms underlying pulmonary injury and vascular dysfunction during *O. tsutsugamushi* infection. First, the timing of ICAM-1 on the surface of lung-derived CD31+CD45- ECs was concurrent with the appearance of signs of vascular injury and decrease in cell junction proteins (Fig 1 and S1 Fig). Since ICAM-1 promotes circulating immune cells to bind to the endothelium and extravasate into inflamed tissues [49], increased ICAM-1 surface expression likely contributed to immune cell influx into the infected mouse lungs (Fig 5, S3 Fig). Our observed *ICAM1* and *IL-8/CXLC8* up-regulation in infected human EC cultures (S1C Fig) were

consistent with other reported studies of scrub typhus patients [50]. It remines unclear as to what specific cytokine networks regulate EC activation *in vivo* during *O. tsutsugamushi* infection; however, it is known that endothelium-activating proinflammatory cytokines, such as TNFα and IL-6, are increased in various mouse tissues in late stages of severe infection, as well as in the serum samples of severe scrub typhus patients [35, 51].

Second, a notable reduction of Tie2 proteins was concurrent with significant Ang2 production and/or release at the severe stages of infection (Figs 2 and 3). To date, there are no reports for Tie2 expression levels in scrub typhus patients or animal models, although our group previously showed increased *ANG2/ANG1* transcript ratios in *O. tsutsugamushi*-infected human EC cultures and mouse tissues [35]. Our findings of significant and progressive reduction in Tie2 protein and its mRNA levels, as well as functional pTie2 level, in the lungs of lethally infection mice are important from basic research and clinical points of view. It is known that the Ang1/Tie2 axis is essential for vascular remodeling and endothelial cell stabilization, as either knockout Ang1 or Tie2 in mice is embryonic lethal [52, 53]. Given the critical function of Tie2 receptor in vascular physiology and integrity, it will be important to further examine whether our observed reduction in Ang1 and Tie2 is due to direct endothelial damage or signaling from nearby pericytes and recruited immune cells, such as activated neutrophils observed in our study (S2 Fig) and in other infection models [54]. Research in these areas would be of great value, as angiopoietin- or Tie2-targeted therapies have been evaluated as alternative treatment strategies for severe sepsis [55, 56], severe dengue [57], and in cerebral malaria [39] infection models to restore endothelial quiescence during infection. Our clinical observation that increased serum Ang2 in human scrub typhus patients correlates with *O. tsutsugamushi*-specific antibody titers demonstrates the utility of Ang2 as a pathogenic biomarker, and highlights the potential of use Ang2- or Tie2-targeted therapies for severe scrub typhus, as in patients with severe sepsis [32, 33] and malaria [58].

Monocytes/MΦs play important roles in infection with *O. tsutsugamushi* and other closely related *Rickettsia* species [22, 59]. Previous findings [22] are consistent with our observation of an increased accumulation of CD64[+]CD11b[+]Ly6G[-] MΦs/monocytes and the close association of IBA-l[+] phagocytes with *Orientia* in the lungs at D6 and D9. Yet, our findings of selective recruitment/expansion of M1-skewed MΦs/monocytes, and the suppression of M2 cell activation, in the infected mouse lungs were particularly novel and important. This is the first *in vivo* evidence for M1-polarized MΦ responses in *O. tsutsugamushi*-infected mouse lungs, which was consistent with previous *in vitro* and *ex vivo* studies for M1-skewed gene transcription programs in *O. tsutsugamushi*-infected human monocytes and MΦs [24, 43]. At present, it is still unclear whether the predominately M1 MΦ population we observed contributes to the killing of *O. tsutsugamushi* and the damage of vascular tissues, as seen for protective versus pathogenic roles in other lung infection models [60]. Since a significant Ang2 release and Tie2 decrease was detected at D6, followed by strong polarization to M1 MΦs (Figs 2–4), it will be important to examine whether endothelial activation precedes the MΦ polarization and contributes to the strong M1 phenotype via the release of endothelial factors, such as Ang2, particularly on Tie2 expressing monocytes/ MΦs [61]. Given the reports that anti-Ang2 antibody treatment during pulmonary bacterial infection can decrease MΦ recruitment and inflammation [29], and that signaling via Tie2 on monocytes/MΦs can promote a proinflammatory profile [62], it will be interesting to determine how Ang2/Ang1 signaling on both endothelial cells and MΦs promote cellular recruitment and activation during *O. tsutsugamushi* infection.

Conditions that promote the killing or growth of *O. tsutsugamushi* remain unclear, in part due to difficulties in bacterial cultivation, genetic modification, or visualization for studying the host-bacterium interactions [63]. The recognition of damage- or pathogen-associated molecules by host receptors can result in MΦ polarization and activation [64]. M1 polarization,

with the production of reactive oxygen and nitrogen species, is well known for killing intracellular bacterial pathogens such as *Listeria monocytogenes*, *Legionella pneumophila*, and *Rickettsia* spp. [65–67]. Our *in vitro* comparative studies revealed limited bacterial replication in LPS-primed M1-like MΦs (Fig 6), supporting a previous report for the role of NOS2-mediated mechanisms in the control of *O. tsutsugamushi* Karp strain [22]. Yet, our *in vitro* findings were contradictory to another reported study, in which NO-enhanced the growth of *O. tsutsugamushi* Ikeda bacteria was observed in LPS-activated RAW 264.7 murine macrophages at days 6 to 8 post-infection [23].

For bone marrow-derived MΦs, we found comparable bacterial loads in LPS-, IL-4-, and non-primed cells at 3 h, implying similar attachment and invasion of the bacteria under these two treatments. However, IL-4-primed M2 and non-primed MΦs contained 10- and 8-fold more *O. tsutsugamushi*, respectively, than LPS-primed M1 cells at 72 h. While it is unclear as to how M1/M2 polarization effects additional mechanisms of *O. tsutsugamushi* clearance in MΦs, such as autophagy, live bacteria are known to efficiently escape macrophage autophagy *in vitro* [68]. The use of transgenic mouse strains for tracking MΦ subsets would also help reveal whether *O. tsutsugamushi* bacteria preferentially replicate within M2 MΦs *in vivo* or contribute to the impairment of type 2 immune responses.

While the mouse model used here allows us to examine host-bacterium interactions and immune alterations, this model has some intrinsic limitations, as it bypasses bacterial dissemination from the local/skin sites. Nevertheless, this model has several advantages over self-limited infection models following the subcutaneous or intradermal inoculation of the bacteria via needles into inbred strains of mice [18, 22], for the examination of innate immune responses in visceral organs. Compared to a lethal infection initiated by feeding *O. tsutsugamushi*-infected mites on outbred mice, a technically challenging model with high variability [69], our model provides more consistent results and lethality, permitting the analysis of a given host molecule using gene-targeted knockouts on the C57BL/6 background. More importantly, our lethal model mimics certain pathological aspects of severe scrub typhus observed in humans, uncovering tissue-specific immune alterations that have never been described previously. For example, our findings of elevated Ang2 proteins in *O. tsutsugamushi*-infected lungs and increased *ANG2* expression in multiple organs [35] are consistent with clinical studies of scrub typhus patients (Fig 2), which supports the potential for monitoring serum Ang2 levels as an indicator of disease severity and treatment outcome.

In summary, this study has revealed new insights into immune dysregulation and pathogenesis of severe scrub typhus. Through comprehensive analyses of *O. tsutsugamushi*-infected mouse lung tissues, we have provided the first evidence for the production of endothelial destabilizing factors and *in vivo* polarization of lung recruited MΦs. Our findings of polarized M1-like responses in the lungs at late stages of disease argue for immune-based restriction of bacterial replication as well as immunopathogenesis. While the molecular mechanisms underlying host-bacterium interaction and immune dysregulation remains unclear at this stage, it is conceivable that serum and tissue Ang2 levels would be a molecular biomarker for severe scrub typhus and a potential therapeutic target for treatment. A better understanding of infection- versus immune-mediated dysregulation will help design treatment strategies for severe scrub typhus cases.

## Materials and methods

### Mouse infection and ethics statement

Female C57BL/6 mice were purchased from Envigo (Huntingdon, United Kingdom), maintained under specific pathogen-free conditions and used at 6–9 weeks of age, following

protocols approved by the Institutional Animal Care and Use Committee (protocols # 9007082B and 1302003) at the University of Texas Medical Branch (UTMB) in Galveston, TX. All mouse infection studies were performed in the ABSL3 facility in the Galveston National Laboratory located at UTMB; all tissue processing and analysis procedures were performed in the BSL3 or BSL2 facilities. All procedures were approved by the Institutional Biosafety Committee, in accordance with Guidelines for Biosafety in Microbiological and Biomedical Laboratories. UTMB operates to comply with the USDA Animal Welfare Act (Public Law 89–544), the Health Research Extension Act of 1985 (Public Law 99–158), the Public Health Service Policy on Humane Care and Use of Laboratory Animals, and the NAS Guide for the Care and Use of Laboratory Animals (ISBN-13). UTMB is a registered Research Facility under the Animal Welfare Act, and has a current assurance on file with the Office of Laboratory Animal Welfare, in compliance with NIH Policy.

*O. tsutsugamushi* Karp strain was used herein; all infection studies were performed with the same bacterial stocks prepared from Vero cell infection, for which infectious organisms were quantified via a qPCR viability assay [17, 70]. Mice were inoculated (i.v.) with ~$1.325 \times 10^6$ viable bacteria (a lethal dose, 200 μl) or PBS and monitored daily for weight loss and signs of disease. In most cases, tissue samples (4–5 mice/group) were collected at 2, 6, and 9 (or 10) days post-infection and inactivated for immediate or subsequent analyses.

Ethical approval for human samples used in this work was granted by the Institutional Review Board of both Seoul National University Hospital (IRB NO 1603-136-751) and Chungnam National University Hospital (IRB NO 2014-12-006). All patients and healthy volunteers provided written informed consent prior to sample collection.

## Immunofluorescence microscopy and quantification

Mouse lung tissues were processed for immunofluorescent staining, as in our previous report [37]. Briefly, 6-μm frozen sections were blocked and incubated with the following rat or rabbit anti-mouse antibodies (1:200, purchased from Abcam, Cambridge, MA, USA, unless specified): anti-ICAM1, anti-Ang1, anti-Ang2 (R&D Systems/Biotechne, McKinley Place NE, Minnesota), anti-Tie2, anti-VE-cadherin (adherence junctions), anti-occludin (epithelial tight junctions), anti-IBA-1 (ionised calcium binding adapter molecule-1, a MΦ marker), anti-MPO (myeloperoxidase, marker for activated neutrophils), anti-CD45 (BD Bioscience, San Jose, CA, USA), or anti-Ang2 (R&D Systems). Staining endothelium in sections was done with FITC-conjugated *Griffonia Simplicifolia* lectin I (1:100, GSL I-B$_4$ lectin, Vector Lab, Burlingame, CA, USA). Bacteria was stained in various sections using rabbit anti-*O. tsutsugamushi* Karp serum (1:500) [17]. Bound antibodies were visualized by using Alexa Fluor 488- or 555-conjugated, goat anti-rat or anti-rabbit IgG (H+L, 1:1,000–1:2,000, Life Technologies, Grand Island, NY, USA). All sections were stained with DAPI (1:5,000, Sigma-Aldrich, St. Louis, MO, USA). Infected sections stained with secondary Abs and DAPI only served as negative controls to optimize staining conditions. For each section, at least 6 low- and 6 high-magnification fields of the lung sections were imaged on a Carl Zeiss Axio Observer fluorescence microscope (Carl Zeiss Microscopy LLC, Thornwood, NY, USA) equipped with Apo-Tome and Zen imaging software. Acquisition settings were identical among samples of different experimental groups. Representative images at each time point are presented. To measure fluorescent intensity of given markers, gray levels across the entire tissue image (low-magnification, 4 independent images per group) were measured via ImageJ under the same parameter setting. Data are presented at mean ± SEM of the group.

For *in vitro* studies, cells were seeded onto coverslips in 24-well plates (Falcon Corning, Corning, NY, USA). At indicated times of infection, slides were washed, fixed with 4% PFA

for 20 min, and permeabilized with Triton X-100 for 15 min. After blocking with 10% BSA/3% goat serum for 1 h, cells were incubated with serum collected from *Orientia*-infected mice (1:1,500) or rabbit anti-IBA-1 (1:250, Abcam) (1:50) at 4°C overnight and then with a secondary Ab: goat anti-mouse Alexa Fluor 488 (Thermo Fisher Scientific, Waltham, MA, USA) or donkey anti-rabbit Alexa Fluor 594 (Invitrogen/Thermo Fisher Scientific) and DAPI (1:1000, Thermo Fisher Scientific). The cover slips were mounted on slides by using an Antifade Mountant solution (Invitrogen/Thermo Fisher Scientific). Images were taken using an Olympus IX51 microscope (Olympus Corporation, Tokyo, Japan).

## Flow cytometry

Equivalent portions of lung tissues were harvested from infected and control mice, minced, and digested with 0.05% collagenase type IV (Gibco/Thermo Fisher Scientific) in Dulbecco's Modified Eagle's Medium (DMEM, Sigma-Aldrich, St. Louis, MO) for 30 mins at 37°C. Minced tissues were loaded into Medicons and homogenized using a BD Mediamachine System (BD Biosciences, Franklin Lakes, NJ). Lung single-cell suspensions were made by passing lung homogenates through 70-μm cell strainers. Spleen homogenates were made by passing tissue through a 70-μm strainer. Lymphocytes were enriched by using Lympholyte-M Cell Separation Media (Burlington, NC). Red blood cells were removed by using Red Cell Lysis Buffer (Sigma-Aldrich). Leukocytes were stained with the Fixable Viability Dye (eFluor 506) (eBioscience/Thermo Fisher Scientific, Walthalm, MA) for live/dead cell staining, blocked with FcγR blocker, and stained with fluorochrome-labeled antibodies (Abs). The following Abs purchased from Thermo Fisher Scientific and BioLegend (San Diego CA): PE-Cy7-anti-CD3ε (145-2C11), Pacific Blue-anti-CD4 (GK1.5), APC-Cy7-anti-CD8a (53–6.7), APC-anti-Ly6G (1A8-Ly6G), APC-anti-CD31 (390), FITC-anti-ICAM-1 (YN1/1.7.4), Pacific Blue-anti-CD45 (30-F11), PE-anti-CD80 (16-10A1), BV421-anti-CD206 (CO68C2), FITC-anti-CD64 (X54-5/7.1), PerCP-Cy5.5-anti-CD11b (M1/70). Cells were fixed in 2% paraformaldehyde overnight at 4°C before cell analysis. Data were collected by a BD LSRFortessa (Becton Dickinson, San Jose, CA) and analyzed using FlowJo software version 8.86 (Tree Star, Ashland, OR). As previously reported for mouse lung tissues [71] CD45$^+$CD31$^-$ and CD45$^-$CD31$^+$ cells were considered hematopoietic cells and endothelial cells by flow cytometry, respectively.

## Western blot

Protein from lung tissues was extracted with a RIPA lysis buffer (Cell Signaling Technology, Danvers, MA) and quantified with BCA Protein Assay kit (Thermo Fisher Scientific). Protein samples (40 μg/lane) were loaded onto 4–20% SDS-PAGE gels (Bio-Rad Laboratories, Hercules, CA, USA) and transferred onto nitrocellulose membranes (Bio-Rad Laboratories). After blocking non-specific binding sites, membranes were respectively incubated with rabbit Abs specific to mouse Tie2 (1:500, Abcam), phospho-Tie2 (1:400, R&D System, USA), and β-actin (1:15000, Novus Biologicals, USA), and an anti-rabbit secondary antibody (SouthernBiotech, Birmingham, AL, USA). After treatment with the Maximum Sensitivity Substrate (Thermo Fisher Scientific) for 1 min, the light signals were captured by Luminescent Image Analyzer (ImageQuant LAS 4000, GE Healthcare Bio-Sciences AB, Sweden). Protein bands were quantified by using image analysis software (ImageJ). Three independent experiments were performed.

## Infection of mouse bone marrow-derived MΦs

Bone marrow cells were collected from mouse femur and tibia and treated with a red cell lysis buffer (Sigma). For MΦ generation, bone marrow cells were grown in DMEM (Gibco) with

10% FBS, penicillin/streptomycin antibiotics, and 40 ng/ml M-CSF (BioLegend) and incubated at 37˚C. Media was changed at day 3, and cells were collected at day 7 and seeded onto 6- or 24-well plates for overnight. MΦs were treated with either 100 ng/ml LPS (for M1 polarization) or 10 ng/ml mouse rIL-4 (for M2 polarization, Peprotech, Rocky Hill, NJ) for 24 h. Cells were then infected with *O. tsutsugamushi* (5 MOI) and centrifuged at 2,000 RPM for 5 min to synchronize infection of the cells.

## Quantitative PCR and reverse transcriptase PCR (qPCR and qRT-PCR)

To determine bacterial loads, bone marrow-derived MΦs were collected at 3, 24, 48, and 72 hpi by using a DNeasy kit (Qiagen) and used for qPCR assays, as previously described [35]. Bacterial loads were normalized to total nanogram (ng) of DNA per μL for the same sample, and data are expressed as the gene copy number of 47-kDa protein per picogram (pg) of DNA. The copy number for the 47-kDa gene was determined by known concentrations of a control plasmid containing single-copy insert of the gene. Gene copy numbers were determined via serial dilution (10-fold) of the control plasmid.

To measure host gene expression, mouse tissues or *in vitro*-infected cells were respectively collected in RNA*Later* (Ambion, Austin, TX) or Trizol solution at 4˚C overnight to inactivate infectious bacteria and stored at -80˚C for subsequent analyses. Total RNA was extracted by using RNeasy mini kits (Qiagen) and digested with RNase-free DNase (Qiagen); cDNA was synthesized with the iScript cDNA synthesis kit (Bio-Rad Laboratories, Hercules, CA). The quantitative RT-PCR (qRT-PCR) assays were performed with iTaq SYBR Green Supermix and a CFX96 Touch Real-Time PCR Detection System (Bio-Rad). PCR assays were denatured for 3 min at 95˚C, followed by 40 cycles of 10s at 95˚C and 30s at 60˚C. Melt-curve analysis was also used to check the specificity of the amplification reaction. Relative abundance of transcripts was calculated by using the $2^{-\Delta\Delta CT}$ method and compared to housekeeping genes glyceraldehyde-3-phosphate dehydrogenase (GAPDH) or β-actin. Primers used in these analyses are listed in S1 Table.

## Human umbilical vein endothelial cell (HUVEC) infection

HUVECs (Cell Application, San Diego, CA) were maintained in complete Prigrow I medium supplemented with 3% heat-inactivated FBS (Applied Biological Materials, Vancouver, Canada) in 5% $CO_2$ at 37˚C. All *in vitro* experiments were performed between cell passages 5 and 7, as described previously [36, 72]. For infection, HUVECs were cultivated in Prigrow I medium with 10% FBS and seeded onto 6-well plates (Corning Inc., Corning, NY). Confluent monolayers were infected with *Orientia* (3 and 10 MOI) for 24 h and compared with uninfected controls.

## Human serum collection and measurement of Ang2 by ELISA

Human serum samples were collected from healthy volunteers (*n* = 8) and scrub typhus patients (*n* = 32) after obtaining informed consent at the Chungnam National University Hospital in Daejeon, South Korea. Scrub typhus diagnosis was confirmed based on clinical symptoms and a positive serology: a 4-fold or greater rise in the titer of paired plasma or single cut-off titer of an IgM antibody ≥ 1:160 by an indirect immunofluorescence antibody assay (IFA) against *O. tsutsugamushi* antigens or passive hemagglutination assay (PHA) during hospital admission. Healthy volunteers had never been previously diagnosed with scrub typhus, and their sera were negative when examined by IFA. Patient plasma samples were classified into four groups based on their IFA titers. IgM titer was recommended as a diagnostic standard of scrub typhus in South Korea by Korean CDC, based on their serosurvey study of Korean

patients [73]. Ang2 concentration was determined by using a commercial ELISA kit (Abcam), according to manufacturer's instructions.

## Human antibody titer measured by IFA

L929 cells infected with three strains of *O. tsutsugamushi* (Boryong, Karp, and Gilliam strains) were harvested, mixed in equal amounts, and used as antigens to measure total IgG titers against *O. tsutsugamushi* via IFA. Briefly, infected L929 cells were harvested, washed with PBS, seeded onto Teflon-coated spot slides, and fixed with cold acetone for 10 min. The slides were stored at -70˚C until use. Two-fold serially diluted (1:40 to 1:1280 in PBS) patient sera was added to the antigen-coated spot on the slide and incubated for 30 min in a wet chamber at room temperature. An Alexa Fluor 488-conjugated goat anti-human IgG (diluted 1:1000 in PBS, Molecular Probes, Waltham, MA, USA) was used as the secondary antibody. The stained slides were examined under an Olympus FV1000 laser scanning confocal microscope (Olympus, Tokyo, Japan). The endpoint titer of IFA was defined as the highest titer showing a fluorescence signal above the background.

## Statistical analysis

Data were presented as mean ± standard errors of the mean (SEM). Differences between individual treatment and control groups were determined by using Student's t test, utilizing Welch's correction when appropriate. One-way ANOVA was used for multiple group comparisons with a Tukey's Post Hoc for comparisons between groups. Statistically significant values are referred to as *, $p < 0.05$; **, $p < 0.01$; ***, $p < 0.001$; and ****, $p < 0.0001$.

## Supporting information

**S1 Fig. Pulmonary pathology progression and reduced expression of occludin tight junction proteins in infected lung tissues.** Female C57BL/6J mice (4–6 mice/group) were inoculated with 1.325 x $10^6$ of *O. tsutsugamushi* Karp strain. At indicated days of infection, equivalent lung portions were collected. (A) Hematoxylin and eosin staining of lung tissues during lethal challenge demonstrating increased cellular infiltration and alveolar thickening as the infection progresses (scale bars = 50 μm). (B) Frozen sections were processed for immunofluorescent staining and co-stained for occludin (cell-cell tight junctions, red), FITC-labeled GSL I-B$_4$ lectins (green, top rows, scale bars = 50 μm), and DAPI (blue). The close-up views of the boxed areas are shown in the lower row (bar = 20 μm). (C) Flow cytometry analysis of viable pulmonary ECs (CD31$^+$CD45$^-$) collected at early (D0) and late (D9) infection. (D) Cultured HUVECs were infected with bacteria at 3 or 10 multiplicity of infection (MOI, 4 samples/group) and analyzed via qRT-PCR for gene expression at 24 h post-infection. Data are presented as relative to GAPDH values. *, $p < 0.05$; **, $p < 0.01$; and ***, $p < 0.001$ compared to PBS controls. Graphs are shown as mean +/- SEM. Flow cytometric and qRT-PCR data were analyzed by using one-way ANOVA with Tukey's Post Hoc. At least 3 independent mouse infection experiments and 2 independent *in vitro* experiments were performed with similar trends; shown are representative data.
(TIF)

**S2 Fig. Neutrophil and EC activation in infected lungs.** Mice were infected, and lung tissues were prepared for immunofluorescent analyses, as *in Fig 1*. Lung frozen sections were co-stained for MPO (green) and Ang2 (red). The low-magnification images (top rows, scale bar = 50 μm) and close-up views of the boxed areas (bottom rows, bar = 20 μm) are shown.
(TIF)

**S3 Fig. Leukocyte and T cell recruitment in the lungs of lethally challenged mice.** Female C57BL/6J mice (3–5 mice per group) were inoculated with 1.325 x $10^6$ of *O. tsutsugamushi* Karp strain. At indicated days of infection, equivalent lung portions were collected and processed for immunofluorescent staining or flow cytometric analysis. (A) Frozen sections were either co-stained for Ang2 (red) and CD45 (a leukocyte marker, green), or Ang2 (green) and CD3 (a T cell marker, red, bars = 50 μm). The percentage and absolute number of CD3$^+$CD4$^+$ T cells (B), as well as CD3$^+$CD8$^+$ T cells (C), were quantified and compared to non-infected controls (*, $p<0.05$; **, $p<0.01$; ****, $p <0.0001$). Graphs are shown as +/- SEM. Flow cytometry groups were analyzed using one-way ANOVA with Tukey's Post Hoc.
(TIF)

**S4 Fig. MΦ infection in mouse lungs and differentiation *in vitro*.** (A) Female C57BL/6J mice (4–6 mice per group) were inoculated with 1.325 x $10^6$ of *O. tsutsugamushi* Karp strain. At days 2 and 10, equivalent lung portions were processed; frozen sections were co-stained for *Orientia* (red), IBA-1 (green, a macrophage marker), and DAPI (blue), showing images in a low-magnification (top rows, scale bar = 50 μm) and close-up views of the boxed areas (bottom rows, bar = 20 μm). (B) Bone marrow-derived MΦs were treated with LPS (100 ng/ml) or rIL-4 (10 ng/ml) for 24 h and analyzed for the expression of indicated markers via flow cytometry. The numbers represent the percentages (%) of gated cells. (C) LPS- and IL-4-primed cells were analyzed by qRT-PCR for the expression of the indicated markers, showing the polarization of MΦ subsets compared with control cells [45, 74] (*, $p < 0.05$; **, $p < 0.01$; and ****, $p < 0.0001$). Data are shown as +/- SEM and were analyzed using one-way ANOVA with Tukey's Post Hoc.
(TIF)

**S1 Table. Real-time PCR primers of human, murine, and bacterial genes.** The primer sequences used in this study (listed in the 5' to 3' direction).
(PDF)

## Acknowledgments

The authors thank Dr. David Walker for generously providing BSL-3 laboratory equipment and space for conducting experiments, Nicole Mendell for continued training and technical help in many aspects of this study, Dr. Juan Olano and the UTMB Immunology Joint Lab Group for providing helpful suggestions, and Dr. Sherry Haller for manuscript editing assistance.

## Author Contributions

**Conceptualization:** Brandon Trent, Lynn Soong.

**Data curation:** Brandon Trent, Yuejin Liang, Yan Xing, Marisol Esqueda, Yang Wei, Nam-Hyuk Cho, Hong-Il Kim, Yeon-Sook Kim, Jiyang Cai, Jinjun Liu, Lynn Soong.

**Formal analysis:** Brandon Trent, Yuejin Liang, Marisol Esqueda, Nam-Hyuk Cho, Yeon-Sook Kim, Jiyang Cai.

**Funding acquisition:** Lynn Soong.

**Investigation:** Brandon Trent, Yuejin Liang, Thomas R. Shelite, Jinjun Liu.

**Methodology:** Brandon Trent, Yang Wei, Thomas R. Shelite, Jiyang Cai.

**Project administration:** Brandon Trent.

**Resources:** Nam-Hyuk Cho, Jiyang Cai, Jiaren Sun, Donald H. Bouyer, Lynn Soong.

**Software:** Jiyang Cai.

**Supervision:** Donald H. Bouyer, Lynn Soong.

**Validation:** Yuejin Liang, Donald H. Bouyer, Jinjun Liu.

**Visualization:** Yuejin Liang, Yan Xing, Jinjun Liu.

**Writing – original draft:** Brandon Trent.

**Writing – review & editing:** Brandon Trent, Yuejin Liang, Thomas R. Shelite, Jiyang Cai, Jiaren Sun, Lynn Soong.

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
