## [Decision Letter · Decision Letter 0]

18 Dec 2019

Dear Dr. Soong:

Thank you very much for submitting your manuscript "Polarized Lung Inflammation and Tie2/Angiopoietin-Mediated Endothelial Dysfunction during Severe Orientia tsutsugamushi Infection" (#PNTD-D-19-01266) for review by PLOS Neglected Tropical Diseases. Your manuscript was fully evaluated at the editorial level and by independent peer reviewers. The reviewers appreciated the attention to an important problem, but raised some substantial concerns about the manuscript as it currently stands. These issues must be addressed before we would be willing to consider a revised version of your study. We cannot, of course, promise publication at that time.

We therefore ask you to modify the manuscript according to the review recommendations before we can consider your manuscript for acceptance. Your revisions should address the specific points made by each reviewer. 

When you are ready to resubmit, please be prepared to upload the following:

(1) A letter containing a detailed list of your responses to the review comments and a description of the changes you have made in the manuscript.

(2) Two versions of the manuscript: one with either highlights or tracked changes denoting where the text has been changed (uploaded as a "Revised Article with Changes Highlighted" file); the other a clean version (uploaded as the article file).

(3) If available, a striking still image (a new image if one is available or an existing one from within your manuscript). If your manuscript is accepted for publication, this image may be featured on our website. Images should ideally be high resolution, eye-catching, single panel images; where one is available, please use 'add file' at the time of resubmission and select 'striking image' as the file type. 

Please provide a short caption, including credits, uploaded as a separate "Other" file. If your image is from someone other than yourself, please ensure that the artist has read and agreed to the terms and conditions of the Creative Commons Attribution License at http://journals.plos.org/plosntds/s/content-license (NOTE: we cannot publish copyrighted images). 

(4) If applicable, we encourage you to add a list of accession numbers/ID numbers for genes and proteins mentioned in the text (these should be listed as a paragraph at the end of the manuscript). You can supply accession numbers for any database, so long as the database is publicly accessible and stable. Examples include LocusLink and SwissProt.

(5) To enhance the reproducibility of your results, we recommend that you deposit your laboratory protocols in protocols.io, where a protocol can be assigned its own identifier (DOI) such that it can be cited independently in the future. For instructions see http://journals.plos.org/plosntds/s/submission-guidelines#loc-methods

While revising your submission, please upload your figure files to the Preflight Analysis and Conversion Engine (PACE) digital diagnostic tool, https://pacev2.apexcovantage.com/ PACE helps ensure that figures meet PLOS requirements. To use PACE, you must first register as a user. Then, login and navigate to the UPLOAD tab, where you will find detailed instructions on how to use the tool. If you encounter any issues or have any questions when using PACE, please email us at figures@plos.org.

We hope to receive your revised manuscript by March 18, 2020. If you anticipate any delay in its return, we ask that you let us know the expected resubmission date by replying to this email.

To submit a revision, go to https://www.editorialmanager.com/pntd/ and log in as an Author. You will see a menu item call Submission Needing Revision. You will find your submission record there. 

Sincerely,

Ulrike Gertrud Munderloh, Ph.D.

Guest Editor

Ana LTO Nascimento

Deputy Editor

Reviewer's Responses to Questions

**Key Review Criteria Required for Acceptance?**

**Methods**

-Are the objectives of the study clearly articulated with a clear testable hypothesis stated?

-Is the study design appropriate to address the stated objectives?

-Is the population clearly described and appropriate for the hypothesis being tested?

-Is the sample size sufficient to ensure adequate power to address the hypothesis being tested?

-Were correct statistical analysis used to support conclusions?

-Are there concerns about ethical or regulatory requirements being met?

Reviewer #1: The research methodology is acceptable. The reviewer's suggestion in included in the result.

Are the objectives of the study clearly articulated with a clear testable hypothesis stated?

-Is the study design appropriate to address the stated objectives?

Yes

-Is the population clearly described and appropriate for the hypothesis being tested?

Yes

-Is the sample size sufficient to ensure adequate power to address the hypothesis being tested?

Yes for animal study

-Were correct statistical analysis used to support conclusions?

Yes

-Are there concerns about ethical or regulatory requirements being met?

No

Reviewer #2: Yes.

Reviewer #3: Yes

**Results**

-Does the analysis presented match the analysis plan?

-Are the results clearly and completely presented?

-Are the figures (Tables, Images) of sufficient quality for clarity?

Reviewer #1: The resulsts of endothelial cell activation and injury are quite good. Only minor revision is suggested.

However major revision in results of macrophages is required. Some more experiments should be performed.

Does the analysis presented match the analysis plan?

Yes

-Are the results clearly and completely presented?

Yes

-Are the figures (Tables, Images) of sufficient quality for clarity?

Yes

Reviewer #2: Yes.

Reviewer #3: Yes

**Conclusions**

-Are the conclusions supported by the data presented?

-Are the limitations of analysis clearly described?

-Do the authors discuss how these data can be helpful to advance our understanding of the topic under study?

-Is public health relevance addressed?

Reviewer #1: -Are the conclusions supported by the data presented?

Yes

-Are the limitations of analysis clearly described?

Yes

-Do the authors discuss how these data can be helpful to advance our understanding of the topic under study?

Yes

-Is public health relevance addressed?

Not applicable

Reviewer #2: Yes.

Reviewer #3: Yes

**Editorial and Data Presentation Modifications?**

Reviewer #1: Major revision.

Reviewer #2: (No Response)

Reviewer #3: (No Response)

**Summary and General Comments**

Reviewer #1: The resulsts of endothelial cell activation and injury are quite good and straightforward. However, the results of macrophages should be improved in order to avoid the false interpretation.

For Fig.6 The results of M0 macrophages (grown in media with neither LPS nor IL-4) must be included to compare with those of M1 and M2 macrophages. If the results of M0 macrophages are comparable to those of M2, the correct interpretation in discussion should be ‘it is possible that M1 macrophages are activated to more effectively kill OT in phagosome before its escape into the cytosol’.

The detection of OT DNA by PCR cannot discriminate between live (in cytosol) and dead organisms (mostly in phagosome). In my experience, dead OT in macrophages can be detected by PCR for a few days. To discriminate live and dead bacteria, it is better to use confocal microscopy to study colocalization of OT and phagosome marker (e.g. LAMP1). OT is still colocalized in phagosome at 1h, but the colocalization of live OT in phagosome will decrease or disappear at 2h. In contrast, dead OT should longer persist in phagosome. Likewise, bacterial viablility dyes may be tried.

If the imaging experiments cannot be performed, the authors should carefully interpret the findings of bacterial load in M0, M1 and M2 macrophages.

In fig.6B Since it was pure macrophage culture, why did you need to stain the cells with anti-IBA-1? 

A phagosome marker will be more meaningful.

According to the upper comments, discussion (line 342-364) should be re-arranged and better summarized. The authors can discuss that TLR signaling and cytokines (e.g. TNF) from M1 macrophages can prime the cells to kill bacteria in phagosomes before cytosolic escape because these functions of M1 are well established in other intracellular pathogen (e.g. Listeria or Mycobacterium). In addition, the killing of cytosolic bacteria at later time by autophagy or other mechanisms by M1 macrophage can be mentioned and included in the discussion.

Throughout the discussion, the authors should have compared these findings with those of other organisms that cause lung pathology such as Rickettsia spp. or Francisella tularensis or virus that causes pneumonia after hematogenous spread. 

Fig.1A Was the picture clear enough to suggest that OT infects and replicates in endothelial cells? If yes, please clearly specify the finding in the text in order to clearly explain the term ‘close association’ (line156-167). On the other hand, the finding in Fig.1D did not explain the discussion line 301 ‘VEGFR on the infected ECs’. Please discuss and cite the references about infection of OT in ECs of visceral organs in mouse models. In addition, I noticed the upregulation of ICAM-1 on uninfected cells in Fig.1A. The role of secreted cytokines such as TNF in the activation of nearby ECs may be discussed to explain this finding.

For Fig.4 page 48, Fig.4A the right flow cytometry gating strategy, the term M1 and M2 should be included in each quadrant in the figure. Few cells are CD80+ CD206+ double positive cells. What are the phenotype of these cells? On figure legend line 898 page 42, M1 should be CD80+CD206-? M2 cells are CD206+, CD80 negative or positive or regardless of CD80 expression?

Fig.4B page48 the word ‘activated’ is absent in the graph title.

The references of mouse M1 and M2 markers must be cited on the first appearance in the text.

Fig.S3B Why did you use CD40 instead of CD80 staining for M1?

Reviewer #2: (No Response)

Reviewer #3: The authors present here an interesting article using a murine model demonstrating that marked Ang2 production, accompanied with severe impairment in the Tie2 functions, are pathogenic mechanisms of severe vascular damage in O. tsutsugamushi infection. Endothelium in infected lungs presented progressive Tie2 malfunction, increased Ang2 and ICAM-1 expression and pro- inflammatory MΦs at the onset of disease and severe stages of infection. The authors observed extensive recruitment and/or activation of M1-like cells, but marked loss and/or suppression of M2-like cells, during the progression of disease. O. tsutsugamushi infection preferentially stimulated pro-inflammatory innate responses in M1-like monocytes/MΦs, which correlate with the onset of vascular damage. IL-4-primed M2 cells contained 10-fold more O. tsutsugamushi than LPS-primed M1 cells at 72 h. Finally the authors highlight the potential for monitoring serum Ang2 levels as an indicator of disease severity and treatment outcome.

This reviewer mas some general comments / suggestions:

Have the authors considered the potential impact of prior conditions such as malnutrition on the endothelium? Could these factors play synergistic roles with the infection? Mice models of malnutrition could help to elucidate this aspect. The authors may want to discuss on this. 

Between those markers of endothelial dysfunction, bioadrenomedullin is of particular interest. Intravascular bioadrenomedullin preserves endothelial integrity. There is an antibody, adrecizumab, which increases intravascular availability of bioadrenomedullin. It could be interesting to profile this marker if the authors have available serum. Nonetheless, absence of this data do not invalidate the interest of this study. 

In the timeline, what is first, endothelial dysfunction or M1 polarization leading to pro-inflammatory responses? This could help to clarify what is the trigger event of lung damage in this infection. 

Do the authors have any data on Neutrophil degranulation proteins, such as MPO, MMPs, lactoferrin, or NGAL? It could be interested to evaluate their association and timing with endothelial dysfunction.

PLOS authors have the option to publish the peer review history of their article (what does this mean?). If published, this will include your full peer review and any attached files.

Reviewer #1: No

Reviewer #2: Yes: Dr. Munegowda C. Koralur

Reviewer #3: No

---

## [Editor Report · Decision Letter 1]

29 Jan 2020

Dear Dr. Soong,

We are pleased to inform you that your manuscript 'Polarized Lung Inflammation and Tie2/Angiopoietin-Mediated Endothelial Dysfunction during Severe Orientia tsutsugamushi Infection' has been provisionally accepted for publication in PLOS Neglected Tropical Diseases.

Before your manuscript can be formally accepted you will need to complete some formatting changes, which you will receive in a follow up email. A member of our team will be in touch within two working days with a set of requests.

Best regards,

Ulrike Gertrud Munderloh, Ph.D.

Guest Editor

Ana LTO Nascimento

Deputy Editor

---

## [Editor Report · Acceptance letter]

26 Feb 2020

Dear Dr. Soong,

We are delighted to inform you that your manuscript, "Polarized Lung Inflammation and Tie2/Angiopoietin-Mediated Endothelial Dysfunction during Severe Orientia tsutsugamushi Infection," has been formally accepted for publication in PLOS Neglected Tropical Diseases.

Best regards,

Serap Aksoy

Editor-in-Chief

Shaden Kamhawi

Editor-in-Chief
